# Adolescent weight management counseling: The effectiveness of an online training program for primary healthcare professionals in Indonesia

Fransisca H. Agung[1]*, Rini Sekartini[2], Nani Sudarsono[3], Aryono Hendarto[2], Retno Asti Werdhani[3], Meita Dhamayanti[4], Retno Pudjiati[5], Lathifah Hanum[5], Affan Naufal[6], Susan M. Sawyer[7]

1 Faculty of Medicine, Universitas Pelita Harapan, Kelapa Dua, Tangerang, Banten, Indonesia, 2 Department of Child Health, Faculty of Medicine Universitas Indonesia, Jakarta, Indonesia, 3 Department of Community Medicine, Faculty of Medicine Universitas Indonesia, Jakarta, Indonesia, 4 Department of Child Health, Faculty of Medicine Universitas Padjajaran, Bandung, West Java, Indonesia, 5 Faculty of Psychology Universitas Indonesia, Kampus UI, Depok, West Java, Indonesia, 6 Balaraja District Hospital, Tangerang, Banten, Indonesia, 7 Centre for Adolescent Health, Royal Children's Hospital, Murdoch Children's Research Institute, and Department of Paediatrics, University of Melbourne, Parkville, VIC, Australia

* Fransisca.agung@uph.edu

**Data Availability Statement:** All relevant data are within the paper and its supporting information files.

## Abstract

### Background

Overweight and obesity are growing public health concerns globally for which innovative prevention and care delivery efforts are required. We recently developed a web-based training program to improve the quality of health professionals' weight management counseling of adolescents in Indonesia. Having previously confirmed its acceptability, this study aimed to measure the effectiveness of the program through a randomized controlled trial.

### Methods

We recruited 64 primary healthcare professionals from 17 provinces across Indonesia who were randomized to participate in a 4-week online training program (intervention group [IG, n = 32] or a waitlist control group [CG, n = 32]). Using active learning approaches, the training program focused on adolescent development, psychosocial assessment, motivational interviewing (MI), and parent engagement. Participants in each arm were asked to record two counseling sessions with adolescents. These were objectively rated by trained psychologists using a validated tool, and also by qualitative assessment of counseling quality. In both groups, the first recorded counseling session occurred before the training. The second recording took place after the training for IG participants, but not for CG participants.

### Results

IG participants demonstrated significant improvements in their knowledge and counseling skills (p<0.001, t-test). This included improvements in introductory remarks, quality of

**Funding:** The research was supported by Universitas Indonesia (Grant number NKB-NKB-1417/UN2.RST/HKP.05.00/2022). The funder had no role in study design, data collection and analysis, decision to publish, or preparation of the manuscript.

psychosocial assessment, and MI skills. There was no change in the extent of parental involvement. The MI training successfully oriented the counseling sessions towards a more collaborative and participatory conversation for supporting behavioral change.

## Conclusion

This novel online training program improved the knowledge and counseling skills of Indonesian primary healthcare professionals. Greater emphasis on engaging parents and more guidance on conducting telehealth counseling may improve parental involvement in future iterations.

## Introduction

Globally, the prevalence of obesity in children and adolescents has tripled in the last four decades [1]. Low- and middle-income countries (LMIC) have experienced a recent, rapid acceleration of obesity in children and adolescents [2]. This is exemplified by Indonesia, a country of 250 million people and over 35 million adolescents aged 10–18 years [3], where the prevalence of overweight or obesity has increased ten times in the past decade [4], even before the anticipated escalation following the COVID-19 pandemic [5].

Overweight at a young age carries a high risk of living with overweight or obesity in adulthood [6], and brings forward the timing of associated 'adult' conditions such as cardiovascular disease and diabetes [7, 8]. The prevalence of these non-communicable diseases has also been increasing in adolescent populations [9]. This is particularly concerning in Indonesia given that cardiovascular and metabolic diseases such as stroke, heart disease, and diabetes are already the leading causes of mortality and morbidity [10].

Adolescence is a critical period for managing obesity due to the dynamic physiological and psychological changes inherent to this period of growth and development [11, 12]. While behavior change is central to clinical approaches to managing obesity at any age [11, 13], adolescence heralds growing individual autonomy and capacity to regulate health-related behaviors such as food choices and physical activity [13]. However, behavior change interventions to address behavior-related health problems are underutilized in healthcare settings and health professionals are poorly prepared to provide adequate counseling to adults who are over their healthiest weight [7, 9], let alone adolescents.

Motivational Interviewing (MI) is a clinical approach for behavior change counseling [14, 15]. It is the most widely used counseling technique in the field of weight management, including for adolescents in primary care [16–18]. MI is a patient-centered technique for exploring patient motivation and strengths for change, which aims to help patients set and reach their own goals [15, 19]. To increase the likelihood of behavior change, it is also important to consider social and physical environments, including the availability and accessibility of resources that support more healthy lifestyles [20]. Furthermore, as parents are critical influencers of social and physical home environments for adolescents [21, 22], their involvement in adolescent weight management is widely recommended [23].

In Indonesia, there is no specific teaching on behavioral counseling in medical [24] and nursing schools [25]. There is similarly an absence of continuing medical education programs to help primary healthcare providers develop the necessary attitudes, knowledge and skills to address behavior change in adolescents who are over their healthiest weight. To provide accessible training across a huge archipelago like Indonesia, and in the context of COVID-19, we

recently developed an internet-based training program to promote behavior change counseling for weight management in adolescents. Previously, the content and mode of delivery of the training pilot was rated highly by participants [26]. The aim of the current paper is to assess the effectiveness of this training program through a randomized controlled trial of primary healthcare professionals to understand its feasibility and identify areas of refinement.

## Methods

The study design was a two-arm randomized controlled trial that recruited 64 health professionals from primary healthcare centers in 17 of Indonesia's 34 provinces. This study was conducted according to the Declaration of Helsinki and conformed with the ethical standards of the Ethics Committee of Fakultas Kedokteran Universitas Indonesia (approval number 829a/UN2.F1/ETIK/PPM.00.02/2021). The study was also undertaken with formal permission from the Indonesian Ministry of Health (MoH). The participating provinces were identified by the MoH, based on the availability of adolescent-trained primary care staff.

Following recruitment to the study, participants (primary healthcare doctors and nurses) were randomized to receive the training (intervention group [IG] or waitlist control group [CG]) based on their district. At baseline and post-intervention, a pre- and post-training knowledge test was administered and two audio-recorded counseling sessions with adolescents with overweight or obesity and their parents were submitted by participants for assessment. The primary outcome of the training was change in health professional knowledge of behavior change counseling, while the secondary outcome was change in counseling skills, using objective ratings of each audio-recording that were made by trained professional observers (clinical psychologists) using a specially designed assessment measure. Following submission of their pre-training audio-recording, the IG participated in the 4-week training program, and then audio-recorded a second counseling session with the same two adolescents. The CG participants submitted their second audio-recording at the same time as the IG, prior to them undertaking the training (Fig 1).

### Recruitment of study participants (health professionals) and adolescents

The sample size was determined using the mean-difference formula for RCTs on motivational interviewing for obesity [27]. Based on this calculation, 60 adolescents were required for each group to assess counselling effectiveness [27]. We undertook a brief survey of health professionals who considered that they could recruit up to two adolescents each, suggesting that we needed at least 30 health professionals for each group (IG, n = 32; waitlist CG, n = 32). Healthcare professionals who had routinely provided adolescent health services at a primary health care center for at least 6 months were recruited between February 1st–February 15th 2022. Participants, nominated by their provincial health offices, were mostly midwives, nurses and general practitioners. Of these, we selected only the nurses and general practitioners as adolescent nutrition lies within their competencies, but not for midwives. Participants verbally provided informed consent by phone which was followed by them providing written informed consent through the website. These participants completed a baseline survey about their sociodemographic characteristics, personal nutritional status and eating habits, level of physical activities and their own parenting style (for those who were parents). The training was developed using the Walsh and McPhee System Model for clinical preventive care that describes the importance of self-efficacy of health professionals in their counselling on behavioral change [28]. This model recognizes that self-efficacy to counsel patients around healthy habits is influenced by health professionals' own health status and behaviors [28, 29]. Each participant was then required to identify two adolescents with BMI >85th percentile (WHO, 2007). Participants

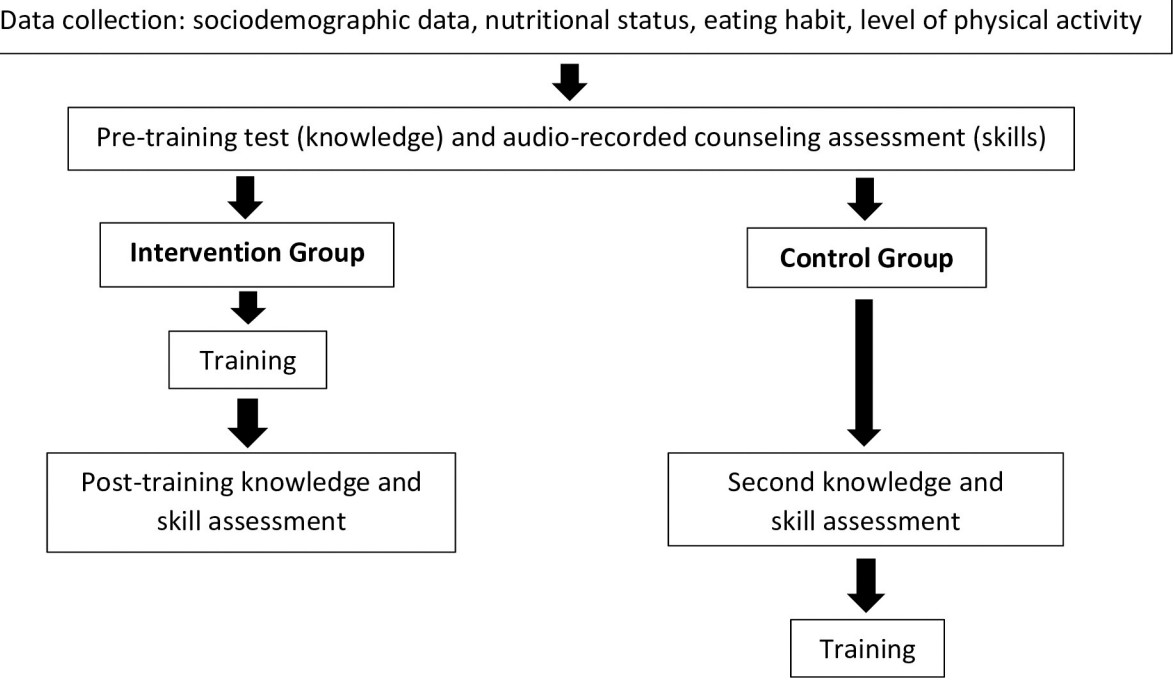

**Fig 1. Study design.**

were asked to explain the study to each adolescent and their parent/s and gain their verbal consent to participate in the study. Prior to the audio-recording of counseling sessions, written informed consent was obtained from both adolescents and parent/s. All resources were available through a website that was specially developed for the study (ramahremaja.id®) which was also used to upload materials (i.e., signed consent forms) and assessments. The resources within the website were accessible once participants had created an account through the website and had registered for the training.

## The training program

The training program was internet-based and consisted of synchronous and non-synchronous sessions (blended learning) that were completed over four weeks across March to April 2022. Following a needs assessment [30], the training program was developed using constructive alignment theory to ensure the alignment between the intended learning outcomes, learning activities and assessment tasks [31]. It consisted of four main topics: adolescent growth and development, parenting adolescents, healthy eating and physical activity, and behavioral change counseling techniques for adolescent patients using MI approaches. Each topic consisted of individual learning tasks and an online meeting with all participants. Active learning principles underpinned the development of all learning activities, which utilized multifaceted materials including explanatory videos, audio recordings of exemplar consultations with adolescents, quizzes, virtual interactive case vignettes, and written assignments (e.g., develop a healthy meal plan, develop a schedule for physical activity, map local healthy lifestyle resources, develop a personal behavioral change plan). The agenda for each weekly online meeting consisted of group work, a group presentation and an interactive discussion lead by the facilitator. In the final online meeting, participants engaged in a formal role-play with adolescents in a

small group setting. Constructive feedback was provided to each participant, with each member of the group learning from feedback to others

## Knowledge assessment

The primary outcome was improved health professional knowledge of behavior-change counseling for weight management of adolescents. This was assessed using a 47-item questionnaire that was specifically developed for the study that consisted of questions exploring knowledge of healthy eating, physical activity, screen time, basic communication skills with adolescent patients, psychosocial screening, motivational interviewing techniques and parent roles in adolescent behavior change. Content validity was determined qualitatively by nine experts (clinicians, academics, members of relevant professional organizations and representatives of the Ministry of Health) to ensure that the content was consistent with current national guidelines [32–34]. Following this, face validity was undertaken by 11 national trainers of adolescent friendly health services in Indonesia. Item analysis was done by calculating the Item Difficulty Index (IDI) using the average score of each item and Discrimination Index (DI) using Point of Biserial coefficient correlation in a sample of 67 health workers from 30 out of 34 provinces in Indonesia [35]. The item analysis procedure was followed by consideration of the learning objectives and materials and led to revision of some items. The final knowledge assessment tool had a Cronbach-α 0.786, indicating good internal consistency.

## Counseling skill assessment

The secondary outcome was improved counseling skills for weight management of adolescent patients. Assessment of counseling skills was evaluated both quantitatively and qualitatively by trained professional raters, as well as by participant self-assessment.

## Quantitative assessment

Assessment tools were specially developed for this study, based on the content of the training module. With the plan of using trained professional raters (psychologists), the assessment tool was modified from the Behavioral Change Counseling Index (BECCI), a commonly used tool for evaluating behavioral-change counseling quality in healthcare settings [36, 37]. Three further items were added which were specific for counseling sessions with adolescent patients, namely: greeting adolescents and their parents; inviting adolescents to talk on their own without their parents (including a confidentiality statement) [38]; and, discussing the approach to parental involvement with the parents. The scoring for each item in the assessment tool was also modified from the original BECCI. Instead of a 5-point Likert scale, our assessment tool used a 4-point Likert scale with slightly modified explanations of each item with the intention of improving its objectivity [39, 40]. A rating of 0 was the lowest (no statements heard) with 3 being the highest (all required statements heard). The assessment tool was called the Adolescent Behavioral Change-Counseling Assessment Tool (ABC-CAT) [41].

   The ABC-CAT underwent a validation process. Firstly, content validity was undertaken by nine experts, followed by assessment of face validity by ten clinical psychologists. Construct validity was undertaken with corrected correlation coefficient analysis (n = 125) using the first counseling session review from both groups (IG and CG). The inter-item corrected value was more than 0.3 for all items and Cronbach-α 0.839, which revealed good validity and good internal consistency [41].

   The final ABC-CAT [41] assessed four domains: 1) Introduction (a greeting and how to start the counseling session); 2) Psychosocial screening (this commenced by inviting the adolescent to talk without their parents, included a confidentiality statement prior to the

psychosocial screening which focused on evaluating the social determinants of healthy weight); 3) MI basic communication skills and MI stages (engaging, focusing, evoking and planning); and 4) Parental involvement (this evaluated whether the participants discussed any of two parenting roles, namely around structuring and supporting behavior change in adolescents).

Eight trained clinical psychologists assessed the audio-taped counseling sessions. These psychologists underwent training for the assessment, after which they each scored 15 audio recordings from an earlier pilot for the study. The inter-rater reliability score (the inter-class correlation coefficient test using the simplified Winer and Walter formula) was 0.942 (CI 0.868–0.973), which showed strong agreement [42].

Each participant uploaded two sets of two recorded counseling sessions onto the website, four weeks apart. IG participants uploaded an audio-recording of the same two adolescents before and after the training. The CG participants also uploaded two audio-recordings on two occasions of the same two adolescents, both prior to training. Each recording was coded and assigned to a rater by an embedded system in the website. The system ensured that each of the four recordings submitted by each participant was reviewed by different raters, who reviewed them blindly.

## Qualitative assessment

Qualitative assessment is a flexible, holistic and nonstatistical assessment method where participants can process what they learned from the experience by provision of feedback soon after the counseling session [43, 44].

For each audio-recording, the trained rater was asked to write a narrative description about the strengths and weaknesses of each counseling session and to provide an overall assessment of its quality. This narrative description was made available to each participant, confidentially, through their training portal on the website.

## Data analysis

A summary of baseline participant data was developed to show the similarities between the two groups. To evaluate the effectiveness of the training, we compared changes in knowledge and skills between each group. Differences between pre- and post-intervention knowledge scores were assessed by group. Change in skills was assessed quantitatively using differences in the ABC-CAT assessment questionnaire, and also qualitatively. Continuous variables were described by standard descriptive statistics containing mean and standard deviation. Comparison between groups was done using the unpaired t-test.

## Results

### Sample characteristics

A consort diagram that details recruitment and retention is shown in Fig 2.

A total of 64 health professionals (52 females) were recruited, which comprised 35 nurses and 29 physicians. Fourteen (22%) reported some prior training in adolescent friendly health services (AFHS). The adolescent patients who consented to be audiotaped were similar in age and sex by group. Each adolescent was required to be accompanied by one of their parents. The parents were predominantly mothers, although there were 17 fathers in each group. As show in Table 1, around two-thirds of health professionals and parents themselves experienced overweight or obesity. The predominant weight category for adolescents was obesity rather than overweight (IG, 92%: CG, 91%).

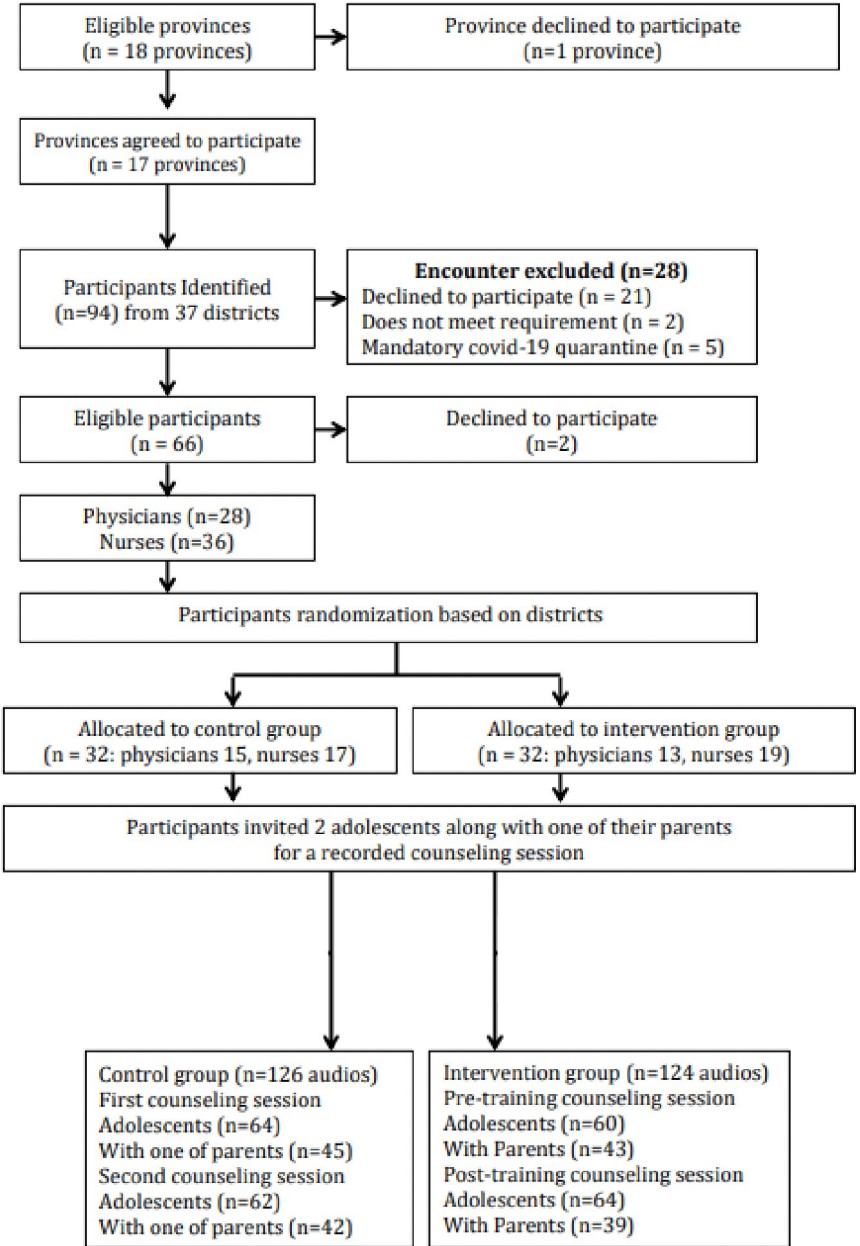

**Fig 2. Consort diagram.**

In both adult populations (health professionals and parents) a proportion of participants had unhealthy eating habits (low fruit consumption, deep fried food, sugary beverages, ultra-processed food and junk meal). Mostly, this was eating vegetables less frequently than recommended and reporting unhealthy snacking. A higher proportion of adolescents than parents reported low consumption of vegetables and high levels of unhealthy snacking. Adolescents also commonly reported a variety of other unhealthy eating behaviors. More than half of the adolescents had low levels of physical activity. For each of these lifestyle characteristics, there were no significant differences between adolescents by group (Table 1).

**Table 1. Nutritional status and lifestyles of health professionals, parents and adolescents.**

| . | Health Professionals | | Parents | | Adolescents | |
|---|---|---|---|---|---|---|
| | **Intervention Group** | **Control Group** | **Intervention Group** | **Control Group** | **Intervention Group** | **Control Group** |
| | **n = 31 (%)\*** | **n = 32 (%)\*** | **n = 46 (%)\*** | **n = 68(%)\*** | **n = 65(%)\*** | **n = 68(%)\*** |
| **Weight status** | | | | | | |
| Normal weight | 11 (35.5) | 13 (40.6) | 17 (37) | 29 (42.6) | 0 (0) | 0 (0) |
| Overweight | 15 (48.3) | 8 (25) | 14 (30.4) | 22 (32.3) | 5 (7.7) | 6 (8.8) |
| Obese | 5 (16.2) | 11 (34.4) | 15 (32.6) | 15 (22.1) | 60 (92.3) | 62 (91.2) |
| **Diet[1]** | | | | | | |
| Diversity of staple foods [2] | | | | | | |
| Yes | 3(9.7) | 3 (9.4) | 14 (30.4) | 34 (50) | 21 (32.3) | 24 (35.3) |
| No | 28(90.3) | 29 (90.6) | 32 (69.6) | 34 (50) | 44 (67.7) | 44 (64.7) |
| Healthy animal-based food [3] | | | | | | |
| Yes | 3 (9.7) | 3 (9.4) | 3 (6.5) | 7 (10.3) | 5 (7.8) | 9 (13.2) |
| No | 28 (90.3) | 28 (90.6) | 43 (93.5) | 61 (89.7) | 60 (92.2) | 59 (86.8) |
| Healthy plant-based food[3] | | | | | | |
| Yes | 5 (16.1) | 5 (15.6) | 3 (6.5) | 7 (10.3) | 5 (7.7) | 5 (7.4) |
| No | 26 (83.9) | 27 (84.4) | 43 (93.5) | 61 (89.7) | 60 (92.3) | 63 (92.6) |
| Vegetables[4] | | | | | | |
| Yes | 17 (54.8) | 21 (65.6) | 26 (56.5) | 39 (57.3) | 9 (13.8) | 7 (10.3) |
| No | 14 (45.2) | 11 (34.4) | 20 (43.5) | 29 (42.7) | 56 (86.2) | 61 (89.7) |
| Fruit [4] | | | | | | |
| Yes | 11 (35.5) | 8 (25) | 11 (23.9) | 18 (26.5) | 13 (20) | 11 (16.1) |
| No | 20 (64.5) | 24 (75) | 35 (76.1) | 50 (73.5) | 52 (80) | 57 (83.9) |
| Healthy snacks [5] | | | | | | |
| Yes | 3 (9.7) | 0 (0) | 3 (6.5) | 4 (5.9) | 0 (0) | 0 (0) |
| No | 28 (90.3) | 32 (100) | 43 (93.5) | 64 (94.1) | 65 (100) | 68 (100) |
| Fast food [6] | | | | | | |
| No | 17 (54.8) | 20 (62.5) | 26 (56.5) | 40 (58.8) | 17 (26.1) | 29 (42.6) |
| Yes | 14 (45.2) | 12 (37.5) | 18 (43.5) | 28 (41.2) | 48 (73.9) | 39 (57.4) |
| Instant food [7] | | | | | | |
| No | 20 (64.5) | 21 (65.6) | 27 (58.7) | 27 (39.7) | 20 (30.8) | 13 (19.1) |
| Yes | 12 (35.5) | 10 (34.4) | 19 (41.3) | 41 (60.3) | 45 (69.2) | 55 (80.9) |
| Sweet drink [8] | | | | | | |
| No | 13 (41.9) | 12 (37.5) | 17 (36.9) | 18 (26.5) | 11 (16.9) | 8 (11.8) |
| Yes | 18 (58.1) | 20 (62.5) | 29 (63.1) | 50 (73.5) | 54 (83.1) | 60 (88.2) |
| Cooking routine at home [9] | | | | | | |
| Yes | 22 (71) | 20 (62.5) | 42 (91.3) | 54 (79.4) | 42 (64.6) | 47 (69.1) |
| No | 9 (29) | 12 (37.5) | 4 (8.7) | 14 (20.6) | 23 (35.4) | 21 (30.9) |
| **Level of Physical activity[10]** | | | | | | |
| Low | 22 (71) | 20 (62.5) | 22 (47.8) | 28 (40) | 36 (55.3) | 37 (54.4) |
| Moderate | 9 (29) | 12 (37.5) | 21 (45.7) | 36 (52) | 25 (38.5) | 23 (33.8) |

(*Continued*)

**Table 1.** (Continued)

| . | Health Professionals | | Parents | | Adolescents | |
|---|---|---|---|---|---|---|
| | Intervention Group | Control Group | Intervention Group | Control Group | Intervention Group | Control Group |
| | n = 31 (%)* | n = 32 (%)* | n = 46 (%)* | n = 68(%)* | n = 65(%)* | n = 68(%)* |
| High | 7 (22) | 8 (25) | 3 (6.5) | 4 (8) | 4 (6.2) | 8 (11.8) |

[1]Using the Food Frequency Questionnaire in the last week

[2]There is staple food native to Indonesia other than rice > 1x per week; [2]Various animal and vegetable side dishes (including fish and red meat, tofu, tempeh / beans) and fried side dishes no more than once per week

[3]Large portions and frequency of fruits and vegetables according to balanced nutrition recommendations

[4]Healthy snacks: no packaged snacks / high sugar / salt more than 1x per week

[5]Fast food no more than 1x/week

[6]Instant noodles or other instant food no more than 1x/week

[7]Sugary drinks no more than 1x per week

[8]Cook 5x or more in last week

[9]Physical activity level based on the short version of the International Physical Activity Questionnaire.

*Despite multiple efforts, only 46 out of 64 parents in IG completed the questionnaire and 1 participant from the IG did not complete the questionnaire. There were participants (1 in IG and 4 in CG) who changed their adolescent patients for the post-training / second counseling and these patients were required to submit the questionnaire as well. Thus, there were 65 adolescents in IG and 68 adolescents in CG, instead of 64.

## Knowledge assessment

At baseline, there were no differences in knowledge scores (p = 0.902, unpaired t-test) between IG and CG participants. Four weeks later, the IG had significantly higher knowledge scores (p < 0.001, unpaired t-test) than the CG (Table 2).

## Counseling skill assessment

**Quantitative assessment (ABC-CAT).** As part of training and assessment, participants engaged in a counseling session with adolescents, which they audio-recorded and uploaded to the study website. Due to the COVID-19 pandemic, participants were free to conduct counseling sessions remotely by video (e.g., zoom, telephone) or face-to-face at the clinic. A total of 256 sessions were uploaded, 128 sessions from each group. Five of 64 adolescents did not return for the 2nd counselling session (IG, n = 1; CG, n = 4); two did not respond to follow-up texts and phone calls, two had moved to another city, and one was hospitalized due to a traffic injury. Instead, audio recordings were made with other adolescents who met the study criteria. A total of 116 (45%) sessions were undertaken virtually, which were similarly distributed by group (IG 47%, CG 44%). At baseline, there were no differences in assessment by group (Table 3). Assessment of the 2nd audio-recording revealed significant differences (p <0.001, unpaired t-test) in the counselling quality of participants in the IG compared to the CG, as measured by the ABC-CAT domains.

Significant differences were seen in relation to both aspects of psychosocial screening, namely the provision of a confidentiality assessment and psychosocial screening itself. There

**Table 2. Comparison of the first and second knowledge tests, by group.**

| Knowledge value | Control Group | Intervention Group | p value (t-test, unpaired group) |
|---|---|---|---|
| | x̄ ± SD | x̄ ± SD | |
| First test (pre-training) | 63.7 ± 11.3 | 64.0 ± 9.9 | 0.902 |
| Second test (post-training for the intervention group) | 67.2 ± 9.9 | 79.1±12.3 | < 0.001 |

**Table 3. Assessment of 1st and 2nd audio-recordings using the ABC-CAT, by group.**

| Scoring Item | | Pre-Training / 1st Counseling | | | Post-Training / 2nd Counseling | | |
|---|---|---|---|---|---|---|---|
| | | Control Group | Intervention Group | *p-value** | Control Group | Intervention Group | *p-value** |
| Introduction | | | | | | | |
| 1 | Greets the adolescent and parent/s | 1.77 ± 1.11 | 1.85 ±0.88 | 0.641 | 1.79 ± 1.06 | 1.69 ±0,95 | 0.609 |
| 2 | Asks 1) the issue / chief complaint and 2) a brief history of complaints (1–3 questions) AND 3) provides introduction to the counseling session | 1.28 ± 0,88 | 1.55 ±0.62 | 0.470 | 1.66 ± 0.95 | 2.29 ±0.64 | < 0.001 |
| Psychosocial Screening | | | | | | | |
| 3 | Asks the adolescent to talk alone without their parent, accompanied by a confidentiality statement | 0.98 ±1.23 | 1.32 ±0.96 | 0.098 | 0.64 ± 0.86 | 1.15 ±1.07 | 0.005 |
| 4 | Undertakes psychosocial screening | 1.36 ±1.23 | 1.55 ±0.11 | 0.373 | 1.75 ± 1.11 | 2.19 ± 0.72 | 0.010 |
| Basic communication skill based on MI | | | | | | | |
| 5 | Open-ended questions | 1.98 ± 0.95 | 2.12 ±0.86 | 0.420 | 2.13 ± 1.01 | 2.63 ± 0.61 | 0.001 |
| 6 | Affirmation | 1.19 ± 1.18 | 1.35 ±1.19 | 0.447 | 1.87 ± 1.15 | 2.44 ± 0.84 | 0.002 |
| 7 | Reflective listening | 1.52 ± 0.99 | 1.77 ±0.85 | 0.134 | 1.92 ± 1.05 | 2.53 ± 0.69 | < 0.001 |
| 8 | Summary | 0.72 ± 092 | 0.90 ±0.91 | 0.273 | 1.46 ± 1.16 | 2.18 ± 0.91 | < 0.001 |
| MI stages | | | | | | | |
| 9 | Engaging | 0.920 ± 0.92 | 0.930 ±0.99 | 0.089 | 1.64 ± 1.11 | 2.37 ± 0.83 | < 0.001 |
| 10 | Focusing | 1.3 ± 1.00 | 1.52 ± 1.00 | 0.090 | 1.43 ± 1.12 | 2.34 ± 0.78 | < 0.001 |
| 11 | Evoking | 0.92 ± 1.12 | 1.05 ± 1.15 | 0.531 | 1.07 ± 1.03 | 2.15 ± 0.88 | < 0.001 |
| 12 | Planning | 1.00 ± 0.89 | 1.00 ± 0.97 | 1 | 1.20 ± 1.12 | 2.31 ± 0.80 | < 0.001 |
| 13 | Provides information with elicit-provide-elicit technique | 1.17 ± 0.52 | 1.33 ± 0.75 | 0.165 | 1.38 ± 0.93 | 1.71 ± 1.01 | 0.061 |
| Parental Involvement | | | | | | | |
| 14 Parental involvement | | 0.98 ± 0.81 | 1.05 ± 0.85 | 0.660 | 0.95 ± 0.81 | 0.84 ± 0.85 | 0.455 |
| **Mean** | | 1.25 ± 0.54 | 1.43 ± 0.55 | **0.074** | 1.49 ± 0.70 | 2.05 ± 0.42 | **< 0.001** |

* t-test, unpaired group

was a highly significant difference between the two groups for all but one assessment item for MI (the elicit-provide-elicit skill just failed to reach significance, p = 0.061). There were no differences between groups in the use of introductory greetings (p = 0.609) or in the extent of parental involvement by group (p = 0.455).

**Qualitative assessment.** At baseline, the narrative assessment of communication techniques by qualified psychologists (who were blinded to both the timing [pre/post] and group) suggested there were substantial opportunities for improvement by participants (health professionals) in both groups. The consultations were commonly stilted, with little evidence of a two-way conversation that actively engaged adolescents in the discussion, let alone showed evidence of goal setting or planning for change.

*The counseling session felt interrogative, so the conversation went one way and tended to be monotonous. It would be better if the adolescent was more actively involved in determining their own changes in eating patterns and activities using the MI technique (Counseling 1, CG, Participant 25, Psychologist G).*

*In some sections of the counseling session, the health worker sounded confused about what else to ask the adolescent. To minimize their confusion, the health worker could reflect more on the adolescent's statements and use this to lead to behavioral changes (Counseling 1, IG, Participant 5, Psychologist E).*

After the training, assessor comments suggest there were differences between participants by group. IG participants used more communication techniques, especially more open questions and reflective listening skills. There was also good appreciation of the different MI stages, with the exception of evoking skills.

*It was apparent that the adolescent patient was fully involved in a two-way discussion regarding changes in her behavior, especially in developing diet and activity plans that the adolescent could do on her own (Counseling 2, IG, Participant 4, Psychologist D).*

*[The participant] conducted an individual session with the adolescent, ensured medical confidentiality and asked open-ended questions that made the adolescent feel comfortable. The health worker applied MI techniques, especially around the engaging process, and involved the adolescent in a two-way discussion, evoking using scaling method and inviting the adolescent to plan for change, again using a two-way conversation (Counseling 2, IG, Participant 31, Psychologist A).*

The preparatory steps required for a psychosocial assessment, such as providing the opportunity for adolescents to talk with the health professional alone and making a confidentiality statement, were less well done in comparison to the behavioral change counseling steps. Further, there was no significant difference (p = 0.455, unpaired t-test) in the involvement of parents by group. IG participants provided greater opportunities for adolescents to share their stories and to develop two-way participatory conversations in relation to developing a behavior-change plan.

## Discussion

To our knowledge, this is the first online training program that has been developed to enhance counseling skills for primary health care professionals caring for adolescents with overweight and obesity in Indonesia, a highly populous low-income country from which health professionals could greatly benefit from effective training interventions. Using a randomized controlled trial that was supported by an online web platform, we demonstrated objective improvements in health professional knowledge of counseling approaches, the primary outcome of the study. Importantly, we also showed significant improvements in counselling skills, the secondary outcome, as shown by improved quality of audiotaped counseling sessions with adolescents. We have previously shown excellent acceptability of the training, with participants rating it highly for both the clarity of content and the enjoyment of training methods [26].

This training was ambitious. In terms of content, up to four broad new concepts were introduced to participants across the four-week program. These related to normal adolescent growth and development to appreciate how patterns of overweight and obesity differ from this; approaches to engaging adolescents, communicating with adolescents and parents, and taking a confidential psychosocial history; normal patterns of adolescent nutrition and physical activity, consideration of meal plans for parents, and assisting families to map local resources; and motivational interviewing to address motivation, physical and social environments influencing adolescent lifestyle [20, 30]. While we intentionally recruited participants from AFHS in Indonesia with the expectation that they would have reasonable knowledge of

at least the first two concepts, only a fifth of participants reported some form of training in adolescent health. It is interesting to reflect on how the results might have differed had the participants started with higher baseline knowledge and skills, such as understanding how to involve parents within consultations. Beyond complex content, novel approaches to delivery were also utilized. Firstly, this related to the nature of blended learning within the online training program which included synchronous and asynchronous pedagogical methods. In addition, as the training was conducted during the COVID-19 pandemic, around 45% of counseling session were done remotely, which many participants (and families) had little familiarity with. Notwithstanding the complexity of concepts and context, the extent of active learning approaches designed to enhance knowledge and skills is likely to have contributed to the positive findings from this study.

Despite recognition of the particular value of e-learning and blended learning for health professionals in LMIC, the lack of high-quality studies from LMIC is also well recognized. In 2015, a systematic review by Al-Shorbaji et al. that examined global e-learning and blended learning methods and their effect on knowledge, skills, attitudes and satisfaction found only five studies had been conducted in LMIC [45]. Disappointingly, a more recent systematic evaluation of e-learning for medical education in LMIC from 2020 found that poor quality studies predominated [46]. For example, of 52 reviewed studies, only four (8%) were randomized controlled trials, and only ten (19%) studies assessed change in skills.

Notwithstanding the novelty of content and the complexity of the setting in the peak of the COVID-19 pandemic, there were significant improvements in intervention participants' knowledge and skills, including MI techniques. These findings are consistent with a study by Antognoli et al. that emphasized that clinician confidence in counseling was enhanced by building skills with the opportunity to practice, engage patients, and receive feedback to hone counseling skills [47], as we also attempted.

We failed to show better parental involvement in developing the behavioral-change plan for their children. Beyond focusing on the individual with overweight, providing parents with the necessary understanding, skills and resources to model more healthy lifestyles for the whole family is a central part of adolescent weight management practices [20, 30]. We do not know the extent to which insignificant differences around engaging parents reflected insufficient focus on parenting within the training program, or whether other factors were responsible, such as parent and professional lack of familiarity with online consultations which may have resulted in reduced engagement of parents during consultations with the health professional participants.

As shown in other online training [48], allowing time for participants to flexibly engage with the learning material is arguably a strength of this training. We also believe that the synchronous weekly meetings which provided a fun, interactive learning environment enabled participants to engage with the facilitators and other participants from across Indonesia, which is likely to have enhanced participant motivation and improved the effectiveness of the training [49, 50]. Health professionals with favorable personal health behaviors are more likely to counsel patients about healthy lifestyles than those with less favorable lifestyle habits and perceived low self-efficacy in nutrition care [51]. The lifestyle characteristics of the health professionals were representative of the Indonesian National Health Survey data on eating behavior and physical activity of the Indonesian population [4]. We hypothesize that the focus on health professionals in relation to their own lifestyles (a hands-on exercise was to develop a personal plan for a meal, physical activity and personal behavior change) may have promoted their self-efficacy in conducting a behavioral change counseling session for adolescents.

Consistent with a systematic review of online learning in LMIC [46], a major challenge for some participants was poor internet connectivity, as well as low internet literacy in a minority

of participants. Following our initial pilot study, efforts were made to reduce the bandwidth required, but other approaches such as gathering participants in a place with good internet connectivity to download the training material and upload assessment tasks may also be required. Offering the interactive skill building components of the four-week program in a more intensive manner, face-to-face, may also obviate this.

A strength of this study is the care that went into the development of the training program. For example, it built on a needs assessment [30] involving adolescent and their parents from different socio-demographic backgrounds that was conducted to generate learning objectives and inform the training materials. Using constructive alignment theory [31], careful attention was made to develop the approach to assessment. A nation-wide study informed the development of the knowledge assessment tool and the ABC-CAT was carefully developed and validated as a measure of skill assessment [41]. Objective rating by multiple trained raters of each counseling session was undertaken blindly, managed through the study website. There are also a number of limitations. The study was undertaken during the COVID-19 pandemic, with many health professionals, patients and parents being novice users of online consultations. Given the importance of parents in providing the social and physical environments for healthy lifestyle development at home [20, 30], the lack of improvement in parent engagement within the counseling sessions suggests further study is required to understand whether this reflected the challenges of the pandemic for parents or perhaps a lack of emphasis within the training program. The many concepts and materials that were required to be learnt in a short period raises questions about how well these practices are able to be embedded within daily consultations, something that also warrants further study. Developing a community of practice is one potential strategy to maintain the group's motivation and provide a space to 'trouble shoot' when participants face roadblocks [52].

## Conclusion

This carefully designed online training program shows promise in increasing the quality of primary care professionals' weight management counseling skills for adolescents. Future research is required to determine the impact of improved counseling skills on the actual behaviors of adolescents and their parents. However, this training program, and the website that it is based on, shows potential as a training strategy that warrants further exploration in Indonesia, given the extent of training needs in this highly populous country.

## Supporting information

**S1 File. Inclusivity-in-global-research-questionnaire.**
(DOCX)

**S2 File. Complete raw data.**
(PDF)

## Acknowledgments

The authors thank the Ministry of Health of the Republic of Indonesia, especially the director and the staff of The Directorate of Nutrition, Child and Maternal Health for their endless support throughout the research journey, and the provincial health offices from the 17 provinces who assisted with the nationwide pilot. The authors also thank: Aryo Sudiro from St. Patrick, the e-learning consultant who developed the website; the Indonesia national AFHS facilitators (Indira Dewi, Rinalco Franky, Ni Made Jendri, Ninin Anggreani, Dian Sharie, Dewi Nur, Risna Yanti, lka Maharani, lndah Ariestanti, Dena Restiana, Asep Zaenal); the clinical

psychologist team (Dian Oriza, Annisa Rahmalia, Utari Krisnamurthi, Sri Wulandari, Fina Dwi Putri, Widya Gunawan, Diajeng Tri, Fadhilah Erynanda, Risky Adinda); and Tan Shot Yen, who significantly contributed to the preparation phase of the training material, during and after the training pilot. Lastly, the authors thank all of the nurses and physicians, adolescents and parents who participated in the pilot intervention

## Author Contributions

**Conceptualization:** Fransisca H. Agung, Rini Sekartini, Nani Sudarsono, Aryono Hendarto, Retno Asti Werdhani, Meita Dhamayanti, Retno Pudjiati, Susan M. Sawyer.

**Data curation:** Fransisca H. Agung, Lathifah Hanum.

**Formal analysis:** Fransisca H. Agung, Lathifah Hanum.

**Funding acquisition:** Retno Asti Werdhani.

**Investigation:** Fransisca H. Agung.

**Methodology:** Fransisca H. Agung, Retno Asti Werdhani, Susan M. Sawyer.

**Project administration:** Affan Naufal.

**Resources:** Susan M. Sawyer.

**Supervision:** Rini Sekartini, Nani Sudarsono, Aryono Hendarto, Meita Dhamayanti, Retno Pudjiati, Susan M. Sawyer.

**Validation:** Lathifah Hanum, Susan M. Sawyer.

**Writing – original draft:** Fransisca H. Agung, Affan Naufal, Susan M. Sawyer.

**Writing – review & editing:** Fransisca H. Agung, Rini Sekartini, Nani Sudarsono, Aryono Hendarto, Retno Asti Werdhani, Meita Dhamayanti, Retno Pudjiati, Lathifah Hanum, Susan M. Sawyer.

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
