## [Decision Letter · Decision Letter 0]

10 Jul 2024

PONE-D-23-35751Adolescent weight management counseling: the effectiveness of an online training program for primary healthcare professionals in IndonesiaPLOS ONE

Dear Dr. Agung,

Thank you for submitting your manuscript to PLOS ONE. After careful consideration, we feel that it has merit but does not fully meet PLOS ONE’s publication criteria as it currently stands. Therefore, we invite you to submit a revised version of the manuscript that addresses the points raised during the review process.

We look forward to receiving your revised manuscript.

Kind regards,

Ammal Mokhtar Metwally, Ph.D (MD)

Academic Editor

PLOS ONE

“This research was supported by Universitas Indonesia (Grant number NKB-NKB-1417/UN2.RST/HKP.05.00/2022)”

4. We note that your Data Availability Statement is currently as follows: [All relevant data are within the manuscript]

Additional Editor Comments:

The manuscript is interested meanwhile, the reviewers have raised a number of points which we believe would improve the manuscript and may allow a revised version to be published in PLOS one.

Reviewers' comments:

Reviewer's Responses to Questions

**Comments to the Author**

1. Is the manuscript technically sound, and do the data support the conclusions?

Reviewer #1: No

Reviewer #2: Yes

2. Has the statistical analysis been performed appropriately and rigorously? 

Reviewer #1: No

Reviewer #2: Yes

3. Have the authors made all data underlying the findings in their manuscript fully available?

Reviewer #1: Yes

Reviewer #2: Yes

4. Is the manuscript presented in an intelligible fashion and written in standard English?

Reviewer #1: Yes

Reviewer #2: No

5. Review Comments to the Author

Reviewer #1: When you are studying weight management in teenagers, your target group should be exactly teenagers and health professionals should have been trained during the intervention to help you. The text and title do not match. If you wanted to write this article, you should have only focused on the experts. If it is on teenagers, for example, you should have reported how much weight they have lost

Reviewer #2: advice for an overall edit of written English.

Line141-142: the authors mention that the participants completed a baseline survey. Do explain the purpose of finding the nutritional status and eating habits of the participants and how it impacts the study.

Line 147: good to mention who had permission to access the website with resources.

line 176: state which national guideline you are referring to and what is in it. give a reference to the guideline.

In the results section it is worthwhile to describe how the CG faired with the interviews. Currently you describe how the IG group did pre and post training.

6. PLOS authors have the option to publish the peer review history of their article (what does this mean?). If published, this will include your full peer review and any attached files.

Reviewer #1: No

Reviewer #2: No

---

## [Author Response · Author response to Decision Letter 0]

28 Aug 2024

Dear Editors

Thank you for your helpful review and the opportunity to revise the manuscript 

We have responded below to each of the comments and questions and trust this is satisfactory. 

Kind regards,

Dr Fransisca H Agung 

Response to Reviewers 

Journal Requirements 

Response: Thankyou. Revisions have been made throughout. Uploaded files have been renamed.

2. Please include a complete copy of PLOS’ questionnaire on inclusivity in global research in your revised manuscript. Response: We have completed this questionnaire and will upload it as a supplementary file. 

 -

“This research was supported by Universitas Indonesia (Grant number NKB-NKB-1417/UN2.RST/HKP.05.00/2022)”

Response: The researcher undertook this study as part of her doctoral studies, which were funded by a scholarship from Universitas Indonesia. However, beyond usual research supervisory processes, the funder had no role in study design, data collection and analysis, decision to publish, or preparation of the manuscript. We have added the following statement to the cover letter 

“This research was supported by Universitas Indonesia (Grant number NKB-NKB-1417/UN2.RST/HKP.05.00/2022). The funder had no role in study design, data collection and analysis, decision to publish, or preparation of the manuscript.”

4. We note that your Data Availability Statement is currently as follows: [All relevant data are within the manuscript]

Please confirm at this time whether or not your submission contains all raw data required to replicate the results of your study. Authors must share the “minimal data set” for their submission. 

Response: We believe that the summary of the results provides sufficient information for the study to be replicated. We have added a supplementary file of the qualitative interview and the quantitative data as part of the minimal data set. See Supplementary file 2.

Response: Thank you. This has been expanded as requested. For ethical approval and study permission, see page 6 line 126 – 131. For informed consent statement, see page 7 line 157-159 and page 8 line 167-170 

Questions Reviewer 1 Reviewer 2 

1. Is the manuscript technically sound, and do the data support the conclusions? 

Reviewer 1: No 

Author’s response We consider that our conclusion is well supported by the data presented in the Results section. Our main conclusion states “This carefully designed online training program shows promise in increasing the quality of primary care professionals’ weight management counseling skills for adolescents.” This conclusion is based on the result that the training improved the knowledge and counselling skills of the health professionals. 

We did not set out to test if the training intervention made a difference to patient outcomes, which will be the focus of future research. 

Reviewer 2: Yes

Author's response: Thank you 

2. Has the statistical analysis been performed appropriately and rigorously?

Reviewer 1: No 

Author’s response We believe that our statistical analyses were appropriate, as mentioned on page 12 line 260– 267. 

Reviewer 2: Yes 

Author's response: Thank you 

3. Have the authors made all data underlying the findings in their manuscript fully available? Yes Yes 

Author’s response Thank you Thank you 

4. Is the manuscript presented in an intelligible fashion and written in standard English? 

Reviewer 1 Yes 

Author's response: Thank you

Reviewer 2 No 

Author’s response Thank you The paper has been carefully reviewed and a number of minor changes made to enhance the quality of the English. 

Reviewer Comments 

Reviewer 1 

When you are studying weight management in teenagers, your target group should be exactly teenagers and health professionals should have been trained during the intervention to help you. The text and title do not match. If you wanted to write this article, you should have only focused on the experts. If it is on teenagers, for example, you should have reported how much weight they have lost 

Author's response: Our research aim was to study the effectiveness of a training program that set out to improve health professionals’ counselling skills for adolescents. The design is an RCT but it did not set out to measure adolescent weight change, but rather, to establish whether or not the training program was able to enhance the knowledge and counselling skills of health professionals (the participants of the training) in the intervention group compared to the control group. 

The evaluation of counselling skills was based on the quality of the recorded sessions that was undertaken by health professionals for ‘real world’ counselling of their overweight adolescent patients. 

As such, we consider that our title is consistent with the manuscript. No revision.

Reviewer 2 

Advice for an overall edit of written English.

 We would be pleased for the reviewer’s specific suggestions of where improvements are needed. The senior author is a native English speaker, a professor of adolescent health who has written over 380 papers (H-index 91, 47,000 citations), and who has been fully involved in the manuscript from its earliest phase. The manuscript has been carefully reviewed and a number of changes have been made. 

Line141-142: the authors mention that the participants completed a baseline survey. Do explain the purpose of finding the nutritional status and eating habits of the participants and how it impacts the study.

 Thank you. We have provided more explanation in the text. 

In essence, counselling on weight management requires behavioral change counselling skills. As part of the learning method, participants were required to develop their own behavioral change plan (line 162). This strategy was based on the Walsh and McPhee theory of behavioral change in which personal experience of working on behavior change is expected to help create empathy and self-efficacy that facilitates change with others. We have previously described this in our earlier paper describing the details of the development of the training program, which was also published by Plos One, and which we cite https://journals.plos.org/

plosone/article?id=

10.1371/journal.pone.0294986

 Additional text added

Page 8 Line 161-165

The training was developed using the Walsh and McPhee System Model for clinical preventive care that describes the importance of self-efficacy of health professionals in their counselling on behavioural change. This model recognizes that self-efficacy to counsel patients around healthy habits is influenced by health professionals’ own health status and behaviors.

Line 147: good to mention who had permission to access the website with resources. Thank you, we have added this to the manuscript. See revision on 

Page 8 Line 172 - 173

The resources within the website were accessible once participants had created an account through the website and had registered for the training. 

line 176: state which national guideline you are referring to and what is in it. give a reference to the guideline.

 Thank you for this suggestion. We have added these references (page 9 line 201) and in the reference list (numbers 32-34). 

In the results section it is worthwhile to describe how the CG faired with the interviews. Currently you describe how the IG group did pre and post training. Both groups (the control and intervention groups) were evaluated for their knowledge and skills using the same method. This is described in the Results section on line 268-372 (page 12-18)

Again, the qualitative assessment, which was undertaken by trained professional raters was of both groups. See lines 345-348 for an example of the qualitative assessment for the control group (CG).

---

## [Decision Letter · Decision Letter 1]

2 Dec 2024

Adolescent weight management counseling: the effectiveness of an online training program for primary healthcare professionals in Indonesia

PONE-D-23-35751R1

Dear Dr. Agung,

We’re pleased to inform you that your manuscript has been judged scientifically suitable for publication and will be formally accepted for publication once it meets all outstanding technical requirements.

Kind regards,

Ammal Mokhtar Metwally, Ph.D (MD)

Academic Editor

PLOS ONE

Additional Editor Comments (optional):

Reviewers' comments:

Reviewer's Responses to Questions

**Comments to the Author**

1. If the authors have adequately addressed your comments raised in a previous round of review and you feel that this manuscript is now acceptable for publication, you may indicate that here to bypass the “Comments to the Author” section, enter your conflict of interest statement in the “Confidential to Editor” section, and submit your "Accept" recommendation.

Reviewer #3: (No Response)

Reviewer #4: All comments have been addressed

Reviewer #5: All comments have been addressed

2. Is the manuscript technically sound, and do the data support the conclusions?

Reviewer #3: Yes

Reviewer #4: Yes

Reviewer #5: Yes

3. Has the statistical analysis been performed appropriately and rigorously? 

Reviewer #3: Yes

Reviewer #4: Yes

Reviewer #5: Yes

4. Have the authors made all data underlying the findings in their manuscript fully available?

Reviewer #3: Yes

Reviewer #4: Yes

Reviewer #5: Yes

5. Is the manuscript presented in an intelligible fashion and written in standard English?

Reviewer #3: Yes

Reviewer #4: Yes

Reviewer #5: No

6. Review Comments to the Author

Reviewer #3: I think this paper is ready for publication. Method section seems good, findings and discussion section is also up to the mark.

Reviewer #4: Since we are living in the digital world, conducting health research using online system becomes relevant. I found the research very helpful to promote good health for adolescents and their parents, and enhance health professional capacity to perform their jobs effectively.

Reviewer #5: There is a minor mismatch between the scope and Title. This papers investigate general training of professionals instead of analyzing the effect of the training on adolescents (for example, changes in their weight). This may give the readers a wrong impression regarding the main aim of the study.

As for the parental roles relating to adolescent behavior change, which the study claims have been targeted in the intervention, there is no evidence of a positive change post-intervention. This is acknowledged, but insufficiently insofar as fine tuning future training is concerned.

Finally, it should be noted that despite independence of the study and valuing in using active learning and MI techniques, the authors lacked a detailed specification about the content and the structure of online training program (for example, the examples of the scenarios that were applied in the study).

Proofreading is required

7. PLOS authors have the option to publish the peer review history of their article (what does this mean?). If published, this will include your full peer review and any attached files.

Reviewer #3: No

Reviewer #4: No

Reviewer #5: **Yes: **Shagufta Hamid Ali

---

## [Editor Report · Acceptance letter]

21 Jan 2025

PONE-D-23-35751R1 

PLOS ONE

Dear Dr. Agung, 

I'm pleased to inform you that your manuscript has been deemed suitable for publication in PLOS ONE. Congratulations! Your manuscript is now being handed over to our production team.

Kind regards, 

on behalf of

Professor Ammal Mokhtar Metwally 

Academic Editor

PLOS ONE